# Characterization of a Novel Pathogenic Reovirus in Grasshoppers

**DOI:** 10.3390/v14122810

**Published:** 2022-12-16

**Authors:** Yao Xu, Jingyi Jiang, Zhen Wang, Jingjing Tian, Wangpeng Shi, Chuan Cao

**Affiliations:** Department of Entomology, China Agricultural University, No. 2 Yuanmingyuan West Road, Haidian District, Beijing 100193, China

**Keywords:** reovirus, *Locusta migratoria*, pathogenic, mortality, fecundity

## Abstract

Grasshoppers can swarm in the millions and destroy crops over wide areas, posing a major economic threat to agriculture. A wide range of insect-related viruses has recently been reported in the metagenomics of grasshoppers. Here, we identified and isolated a novel reovirus from grasshoppers, named Acrididae reovirus (ARV). The complete genome of ARV was composed of nine dsRNA segments. Phylogenetic analysis revealed that ARV formed a monophyletic lineage with unclassified insect-associated reoviruses and was sufficiently distinct from known genera of *Reoviridae*. ARV could replicate in its host *Locusta migratoria* and result in host death. Lower-dose ARV infection affected ovary development and resulted in a significant reduction in fecundity. The identification and characterization of a novel pathogenic reovirus could potentially promote the development of new biological control agents.

## 1. Introduction

Grasshoppers (Orthoptera: Acridoidea) are herbivorous and polyphagous, with many species occasionally causing devastating threats to crops and pastures [1]. Traditional grasshopper control relies on chemical pesticides, but it is later restricted because of its accumulation in the food chain and its persistence in the environment [2]. In recent years, alternative biological control methods have been proven effective [3]. Biopesticides such as *Metarhizium acridum* and *Paranosema locustae* have been used in the concerted controlling of grasshoppers in China and have successfully prevented migratory locusts from reaching plague proportions [1,4]. *M. acridum* has also been used in East Africa against recent desert locust infestations and used operationally against the migratory locust in East Timor and against red locusts in East Africa [1,5]. Another promising control method is using entomopathogenic viruses, which usually spread horizontally and transmit vertically in grasshopper populations [6]. To date, only *Melanoplus sanguinipes entomopoxvirus* have been used as a microbial insecticide [7,8], and a picornavirus in *Schistocerca americana* have been discovered [9]. Recently, our study using a metagenomic approach revealed that grasshoppers harbor an enormous diversity of DNA and RNA viruses [10].

Reoviridae is a family of double-stranded RNA (dsRNA) viruses with 9–12 linear genome segments. Ninety-seven species of *Reoviridae* are divided into fifteen genera [11]. Members of the *Reoviridae* have a wide host range, including invertebrates, vertebrates, plants, fungi, and protists [11]. Insect-related reoviruses are classified into genera *Cypovirus*, *Idnoreovirus*, *Dinovernavirus*, and *Fijivirus* [12,13]. Reovirus infections can produce chronic disease in Lepidoptera and Hymenoptera and significantly alter the life history traits of the hosts [12]. *Drosophila S virus* is related to the S phenotype in *Drosophila simulans* [14]. *Diadromus pulchellus idnoreovirus 1* plays a role in helping determine the sex and ploidy of the host [15]. Larvae of *Uranotaenia sapphirina* stop feeding as early as 4 days post-infection with cypovirus, resulting in chronic diarrhea and the loss of body size and weight [16]. Infected pupae are frequently reduced in size and the majority of diseased adults are malformed [16]. However, insect-originating reoviruses have not been reported in orthopteran insects.

A reovirus was found in four grasshopper species in our previous study [10]. Here we further isolate and characterize this virus and name it Acrididae reovirus (ARV). By using *Locusta migratoria*, we find that ARV is pathogenic to its host and reduces survival and reproduction. The ability to rear this virus in the lab using live insects and insect cell cultures indicates the potential for mass production. These results indicate that ARV may be a good candidate for the biological control of grasshopper pests. The fact that ARV can replicate in lab-reared locusts and insect cell cultures provides a good system to study the interactions between insect-specific viruses and their hosts.

## 2. Materials and Methods

### 2.1. Sample Collection and Preparation

*Chorthippus albonemus*, a dominant species in the grasslands of Qinghai Province, China was randomly collected by sweep-netting grassland in August 2018 (9.69°–102.29° E, 35.50°–37.31° N, altitude 3197–3457 m). The mitochondrial cytochrome c oxidase subunit I gene (COI) sequences and morphological characteristics were used for classification [10]. Eggs of *L. migratoria* were collected from Yunan Province, China, and hatched with sterile vermiculite regulated at 30 ± 1 °C, 70 ± 5% RH in growth cabinets (PRX-350B-30). Nymphs were reared in cages (16 × 9.5 × 11 cm) in an artificial climate incubator at 30 ± 1 °C and 70 ± 1% RH and a 16:8 h L:D photoperiod. Twenty-five individuals per cage were maintained and fed on wheat seedlings. Adult locusts were transferred into larger cages (60 cm × 50 cm × 60 cm) under the same rearing conditions. All samples in this study were cultured in a dedicated incubator to avoid the infection of other pathogens.

### 2.2. Virus Purification

A contig of reovirus RdRp was identified from the RNAseq libraries of four grasshopper species, including *L. migratoria*, *C. albonemus*, *Dasyhippus barbipes*, and *Oedaleus decorus asiaticus* [10]. In addition, reovirus was the most abundant virus in *C. albonemus* from Qinghai [10]. Forty individuals of *C. albonemus* were thence homogenized in phosphate-buffered saline (PBS, pH 7.4, Solarbio, Beijing, China) and filtered through sample sieves with 0.003 mm aperture. The grinding fluid without tissue debris was spread on top of a discontinuous sucrose gradient (30%, 40%, 50%, 60% *w*/*v*) and ultracentrifuged at 240,000× *g* for 3 h in an SW 41 Ti Swinging-Bucket Rotor (Beckman Coulter, Brea, CA, USA). The visible virus bands were collected, diluted 10-fold with PBS, and ultracentrifuged for 2 h at 240,000× *g* in an SW 41 Ti Swinging-Bucket Rotor. The virus pellet was then suspended in PBS.

We were not able to isolate pure ARV from the virus mix extracted from *C. albonemus* using sucrose density gradient ultracentrifugation. Therefore, *Drosophila melanogaster* S2 cell lines and *Spodoptera frugiperda* Sf9 cell lines were used to generate a single clone of reovirus from the mixed virus extraction. Serial dilution of the virus mix was carried out from 10^−1^ through 10^−6^. S2 cells were seeded with 1 × 10^4^ cells per well in 96-well plates and incubated overnight. The cell culture’s supernatant was carefully removed and replaced with 100 μL of the virus dilution. Then 100 μL of PBS was added as a control for uninfected cells. The inoculum was removed after 2 h followed by adding 150 μL of the fresh medium to each well. Cell cultures were cultured in a serum-free insect cell culture medium (SFX-Insect, HyClone, Logan, UT, USA) with s 1% Penicillin-Streptomycin solution (HyClone, Logan, UT, USA) at 25 °C.

The total RNA of the cell-virus suspension was extracted after incubation at 25 °C for 7 days. Real-time fluorescence quantitative PCR (qPCR) was used to detect reovirus and other viruses. Specific primers for different viruses are listed in our previous study (Appendix A) [10]. Only reovirus-infected cells were kept to further expand. After culturing, virus-infected cells were lysed via multiple freeze–thaw cycles [17]. Cell debris was removed by low-speed centrifugation twice at 3000 rpm for 10 min. The supernatant-containing virus was ultracentrifuged for 2 h at 240,000× *g* in the SW 41 Ti Swinging-Bucket Rotor to concentrate the virions. The purified virus was harvested in PBS and stored at −80 °C. The Sf9 cell line was also used for the reovirus purification attempts as described above.

### 2.3. Viral RNA Extraction

The virus pellet was transferred to TRIzol (Invitrogen, Carlsbad, CA, USA) for total viral RNA extraction according to the manufacturer’s instructions. Reverse transcription was performed with the PrimeScript™RT reagent Kit (Takara, Osaka, Japan) using random hexamers, and PCR amplicons were carried out using the PrimeSTAR Max DNA Polymerase (Takara, Osaka, Japan). To confirm the presence of reovirus, specific primers targeting the RNA-dependent RNA polymerase (RdRp) were designed based on the viral contigs in our published study (Appendix A).

### 2.4. Transmission Electron Microscopy (TEM)

Twenty microliters of viral particles were dispensed on carbon-coated copper grids for 5 min, washed, and negatively stained with 3% phosphotungstic acid (Solarbio, Beijing, China). Virions were imaged by a Hitachi H-7500 transmission electron microscope (Hitachi High-Technologies, Tokyo, Japan).

### 2.5. Reovirus Genome Sequencing

Viral RNA was extracted from purified virions isolated from S2 cell cultures as described above. The RNA was denatured at 95 °C for 3 min to unwind the double strands. Residual Ribosomal RNA (rRNA) was depleted using Ribo-Zero™ kits (Epicentre, Madison, WI, USA) before library construction. Libraries were constructed using a TruSeq total RNA library preparation kit (Illumina, Sandiego, CA, USA) and paired-end (150 bp) sequencing was performed on the Illumina Novaseq 6000 platform (Illumina). Trimming and assembly of sequencing reads, virus discovery, and calculation of virus abundance were carried out as described in a previous study [10]. Reovirus-related contigs were first identified based on the homology of the amino acid sequence to known reoviruses. The contigs with high expression levels but low homology to known reoviruses were then selected as candidate reovirus sequences. To further confirm all segments of reovirus genome, dsRNAs were isolated from the purified virus following the method described by Khabbazi et al. [18]. dsRNAs were analyzed using 2% agarose gel electrophoresis and visualized in C-150 Gel Imager (Azure, Northlake, IL, USA). Specific primers were designed for assembled contigs with high abundance and similar size to the dsRNA segments (Appendix A). Reverse transcription on dsRNA was performed with the PrimeScript™RT reagent Kit (Takara, Japan) using random hexamers. The reverse transcription product of dsRNAs was used as a template to amplify the segments of the reovirus using the PrimeSTAR Max DNA Polymerase (Takara, Osaka, Japan). Nucleic acid sequences of all segments were further determined using Sanger sequencing (Sangon Biotech, Shanghai, China). The annotation of the genome structure and the construction of the phylogenetic tree were carried out using methods described by Xu et al. [10]. The complete sequences of the reovirus genome were deposited in GenBank (OP837503-OP837511).

### 2.6. Virus Quantification

A classical TCID_50_ assay failed to titrate the viral particles because ARV did not induce a cytopathic effect in insect cell cultures. The Quantitative Real-Time PCR (qRT-PCR) methodology had been proven effective in the determination of reovirus stock titers [19,20]. The preparation of RNA standards and the quantitative real-time PCR assay were carried out as described previously [19,20]. Briefly, primers containing the T7 promoter were designed based on conservative RdRp (Appendix A). Reverse-transcription PCR was performed to amplify cDNAs from total RNA extracts of purified reovirus. The cDNAs served as the templates to produce PCR products using primers containing the T7 promoter. PCR products were recovered using a gel recovery kit (Axygen AP-GX-250) and then used as the DNA template for synthesizing RNA standards. The single-stranded transcribed RNA standards were synthesized in vitro with the DNA template according to the RiboMAX large-scale RNA production system-T7 (Promega, Madison, WI, USA). The RNA standards were reverse transcribed and serially diluted 10-fold to generate standard curves for quantitative real-time PCR. The concentration of cDNA for RNA standards was quantified by the NanoDrop system (Thermo, Waltham, MA, USA) and converted to molecular copies by using the formula described by Escaffre et al. [20].

### 2.7. Infectivity Assays

*Locust migratoria* was a natural host of reovirus and was easy to raise in the laboratory. Newly molted 3rd-instar *L. migratoria* nymphs were therefore injected with 9 × 10^12^ virions into the membrane between the third and fourth abdominal segments using a Nanoinjector (Drummond Scientific, Broomal, PA, USA). An equal volume of sterile PBS was injected as the control. The survival of inoculated locusts was recorded daily until no deaths occurred. The qPCR method was used to demonstrate the successful infection of the reovirus in locusts. Six replicates with twenty-five nymphs for each inoculation were generated.

Furthermore, 3rd-instar locust nymphs were then injected with a lower dose at 9 × 10^10^ virions to increase nymph survival for other pathogenic effects characterization. Guts, ovaries, and heads of infected samples were dissected at 9 time points across all developmental stages, including 1, 3, 5, 7, 9, 13, 17, 21, 24, and 27 days post-infection. RT-PCR was carried out on RNA extracted from guts, ovaries, and heads of infected and uninfected locusts. qPCR was used to test for viral titers. The viral load of a mixture of fifteen individuals was quantified at each time point. Experiments were repeated five times. The infection of the ovaries by the reovirus may affect the reproduction and vertical transmission of locusts, so ovary sizes were measured and compared as an indication of reproduction ability. Moreover, the egg-laying amount of fifteen pairs of locusts was recorded, and the experiments were repeated four times.

To reveal the transmission mode of ARV, twenty-five 3rd-instar nymphs were exposed to the feces of virus-infected locusts for 2 h, and the viral load of the locusts was measured after 24 h. Additionally, eggs laid by infected adults were collected. Total RNA was extracted from two types of eggs, including those that had not been cleaned and had been disinfected with sodium hypochlorite. Viral loads were also measured for nymphs hatched from cleaned or uncleaned eggs. qPCR was used to test for viral titers of eggs. Experiments were repeated 5 times.

### 2.8. Statistical Analysis

To compare the qualitative and quantitative variables between groups, Student’s T-test or the Mann–Whitney U test was used in a univariate analysis. The results were considered statistically significant at *p* < 0.05; specific *p*-values were provided in the text. Statistical analysis was performed using IBM SPSS Statistics 26.0 (Chicago, IL, USA) and GraphPad Prism 8.0.1 (La Jolla, CA, USA).

## 3. Results

### 3.1. A Novel Reovirus Identified in Grasshoppers

RdRp sequences of the reovirus can be successfully amplified from field-collected samples from these four species. Reovirus RdRp of grasshoppers only shared a 30% amino acid (aa) identity with its most closely related RdRp sequence of Hubei reo-like virus 1. A reovirus of grasshoppers can be considered a novel insect-specific virus based on ICTV species demarcation criteria (https://talk.ictvonline.org/ictv-reports, accessed on 1 June 2022). Considering the host origin of the virus, this novel reovirus was named Acrididae reovirus (ARV).

To characterize the phylogenetic relationship of ARV among *Reoviridae*, 38 available RdRp sequences from 15 known genera and unclassified species in the family *Reoviridae* were obtained from NCBI. A phylogenetic tree was constructed using amino acid sequences of RdRp. Phylogenetic analysis indicated that the newly discovered ARV grouped within the subfamily Spinareoviridae but could not be easily incorporated into current *Reoviridae* classifications (Figure 1). The phylogenetic inference revealed a separate clade formed by ARV and other insect-related viruses found in fruit flies, coleoptera, and odonata insects (Figure 1). Notably, this new insect-specific clade was closer to the genus *Fijivirus*, which infected plants, than other insect-related genera such as *Cypovirus*, *Idnoreovirus*, and *Dinovernavirus* (Figure 1). Comparison of RdRp sequences showed that the amino acid sequences of members of this separate clade were less than 30% identical to known genera (Appendix A). Therefore, we speculated that there was a new genus of *Reoviridae*.

### 3.2. Characterization of ARV Virions

Mixed viruses extracted from *C. albonemus* were serially diluted and inoculated into Sf9 and S2 cells to establish a single clone infection. A single clone of ARV was further cultured for 7 days to produce pure ARV. ARV virions were harvested from the cell culture and morphologically examined using TEM. ARV virions were similar in size to reported reoviruses, with an overall diameter of 40–50 nm (Figure 2).

### 3.3. ARV Genome Sequence

First, 2.17 Gb RNA-seq data were obtained for purified ARV virions. De novo assembly generated 1151 contigs ranging from 200 to 9865 nt. Four reovirus-related contigs encoded for RdRp, structural proteins, major core proteins, and minor core structural proteins were discovered based on the homology of the amino acid sequence to known reoviruses. However, running dsRNAs of ARV virions on 2% agarose gel indicated nine genomic bands (Figure 3A). We selected assembled contigs with high abundance and similar size to the dsRNA segments for further verification. PCR amplification and sanger sequencing was used to confirm five other segments that belonged to the ARV genome. These results showed that the complete genome of ARV comprised nine segments, including S1 with 4515 nt, S2 with 4190 nt, S3 with 3886 nt, S4 with 3321 nt, S5 with 1975 nt, S6 with 1959 nt, S7 with 1520 nt, S8 with 1338 nt, and S9 with 1183 nt (Figure 3B). All genomic segments of ARV shared a conserved hexanucleotide terminus at their 3 ends (CGCGAC-3′), which was different from previously reported reoviruses.

### 3.4. Pathogenic Effect of ARV on L. migratoria

The Quantitative Real-Time PCR method was used to quantify the genomic RNA level of purified ARV virions [19]. The transcribed RNA standards were successfully synthesized using a specific primer of viral RdRp (Appendix A). We obtained a standard curve between the threshold cycle (Ct) values and 10-fold serially diluted RNA standards in the range of 10^16^–10^20^ copies/reaction. The standard curve had a reaction efficiency (E) of 94% and a correlation coefficient (R^2^) of 1. The viral genomic RNA level in purified virions and various tissues of infected nymphs were further quantified through the extrapolation of Ct-values to the standard curve in the real-time PCR analysis.

High-dose (9 × 10^12^ virions) ARV has a severe pathogenic effect on *L. migratoria*. ARV-infected 3rd instar nymphs started to die at 3 days post-infection (dpi). The death rate was highest within 3–8 dpi, and viral loads reached a peak (5.99 × 10^18^ virions/μL). The final survival rate for the infected locusts was 19% (Figure 4A), and only a few individuals reached the adult stage.

A lower dose (9 × 10^10^ virions) of ARV, which would not cause early mortality, was injected into 3^rd^ instar locusts to characterize virus replication and pathology in different tissues. ARV was detected in the guts, heads, and ovaries of infected locusts (Figure 4B). Similar to other reoviruses, higher replication of ARV was detected in the guts [21]. The highest viral load was observed in the guts at 3 dpi, followed by a gradual decrease to a stable lower titer after 21 dpi (Figure 4B). However, the viral load of heads and ovaries reached a maximum slightly later, at 5 dpi.

We observed an obvious pathogenic effect of ARV on the development of ovaries. The size of the ovaries of infected locusts was significantly reduced (Figure 5A). The sizes of the ovaries of uninfected adults were 1.138 × 0.528 cm and 0.932 × 0.611 cm at 4 and 7 dpi, respectively, while the ovaries of infected adults at 4 and 7 dpi were 1.026 × 0.300 cm and 0.723 × 0.322 cm (Figure 5B). The replication of ARV in infected locusts produced a characteristic porcelain-white color for the ovaries, which may result from the decrease in vitellin in the ovaries (Figure 5A). Moreover, the number of eggs laid by infected locusts was significantly lower than that of uninfected locusts (Figure 5C).

In addition, ARV can be detected from the feces and from nymphs exposed to the feces of infected nymphs (Table 1). This result indicated that ARV could be horizontally transmitted by the feces of infected locusts. ARV can also be detected on the surface of eggs and in nymphs that hatched from these eggs (Table 1). However, provided the egg surface was disinfected using a compound sodium hypochlorite disinfectant, no transovarial transmission was observed (Table 1). ARV was not detected in nymphs that were hatched from sodium-hypochlorite-treated eggs. These results suggested potential transovarial transmission via the egg surface.

## 4. Discussion

The current known hosts of insect-specific reoviruses are almost exclusively limited to lepidoptera, hymenoptera, and diptera insects [11]. *Cytoplasmic polyhedrosis viruses* (CPVs, also known as Cypoviruses) have been studied for their potential as biopesticides due to their induction of sublethal effects in infected lepidopterans, including lower weight and shorter adult lifespan [22,23]. *Diadromus pulchellus idnoreovirus 1* can interact with ascovirus to down-regulate host immunity for successful parasitism by inhibiting the melanization of the *D. pulchellus* [15,24]. To date, no reovirus has been reported and isolated in orthoptera hosts. Here, we isolate and describe a reovirus, named Acrididae reovirus (ARV), which can infect multiple grasshopper species. Inoculation experiments show that ARV has detrimental effects on the lifespan and reproduction of *L. migratoria*. ARV can be successfully cultured in insect cell cultures and live locusts, which makes it a good candidate to be developed as a biocontrol agent. This is of particular interest for the increasing demand for environmentally friendly novel agents to manage the frequent outbreaks of locust plagues worldwide.

ARV contains only nine dsRNA segments. Additionally, we reveal a conserved hexanucleotide terminus at the 3′ ends that is different from other reoviruses. The conserved terminal sequences of reoviruses play important roles in RNA replication, stability, and immune recognition [25,26]. Phylogenetic analysis indicates that ARV and other reoviruses found in fruit flies, coleoptera, and odanate insects cannot be placed in classified genera of *Reoviridae*. According to current ICTV criteria, the identities of the RdRp amino acid sequence between different genera are generally less than 30%. Therefore, ARV with other unclassified insect reoviruses appears to be a novel genus of *Reoviridae*. Furthermore, they form a distinct clade that closely groups with plant-infecting fijiviruses. This discovery will contribute to deciphering the evolution of *Reoviridae*.

The study of virus pathogenicity and its interaction with hosts is often limited by the difficulty of virus isolation and culture. For example, *Diaphorina citri reovirus* (DcRV) was recently identified from *Diaphorina citri* populations and considered a novel species of *Reoviridae* [27]. Notably, DcRV-encoded non-structural protein P10 is associated with virus spread and biparental transmission to progeny [28]. *Bombyx mori cytoplasmic polyhedrosis virus* (BmCPV) is a relatively well-studied reovirus. A recent study discovers that ROS induced by vSP27 can suppress BmCPV infection via the NF-κB signaling pathway. vSP27 is translated from a BmCPV-derived circular RNA (circRNA-vSP27) [29]. Fortunately, we are able to generate a single reovirus clone, which actively replicated in insect cells and locusts. It provides an excellent laboratory replication model for studying virus–host interactions.

To determine the optimal proposal to apply virus agents in the field, it is important to acquire the most knowledge concerning virus replication dynamics in the hosts. We show that high-dose ARV significantly reduced the survival of infected *L. migratoria*. A lower dose of ARV can jeopardize ovarian development, thus reducing the reproduction of infected hosts. Both features are useful in controlling locust populations in the field. More importantly, given it is found in multiple grasshopper species, it may have wider usage as a biocontrol agent to manage orthopteran pests.

One aspect of ARV replication dynamics that deserves attention is the maintenance of the virus in the host population, indicating persistent infection. Persistent infection is a consequence of interactions between the viral modulation of replication and trade-offs of host immunity [30,31]. Therefore, the isolation and characterization of a novel grasshopper reovirus, ARV, provide a new system to study host–virus interactions.

## 5. Conclusions

In this work, we identified and isolated a novel reovirus from grasshoppers, named Acrididae reovirus (ARV). ARV can be successfully cultured in insect cell cultures and live locusts. Using genome sequencing, our results showed that the complete genome of ARV comprised nine segments. All genomic segments of ARV shared a conserved hexanucleotide terminus at their 3′ ends (CGCGAC-3′), which was different from previously reported reoviruses. Importantly, ARV was pathogenic to its host by reducing host survival and reproduction. These results indicated that ARV can be a good candidate for the biological control of grasshopper pests.

## Figures and Tables

**Figure 1 viruses-14-02810-f001:**
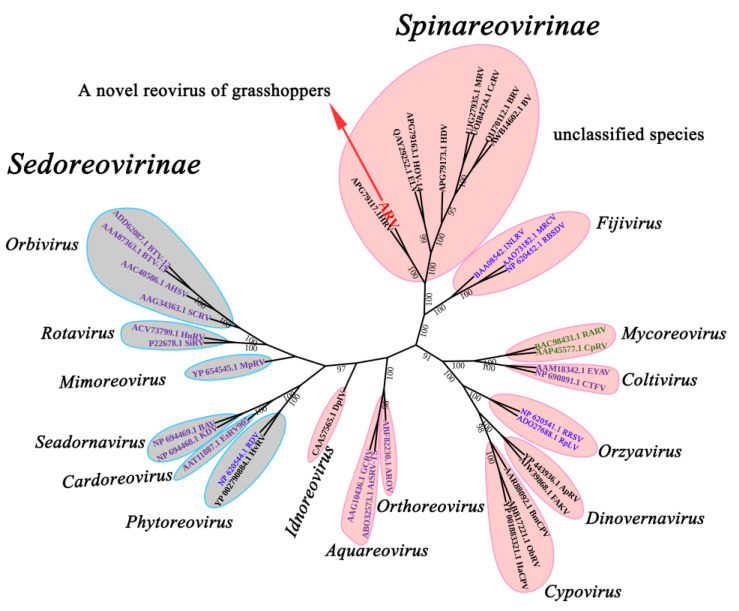
Maximum likelihood phylogeny of *Reoviridae*. Phylogenetic tree was constructed using RdRp sequences. ARV is indicated by red arrow. The nine genera of the subfamily Spinareovirinae are marked in pink, and six genera of the subfamily Sedoreovirinae are marked in grey. Viruses were colored differently according to their hosts: Insects: Black; Plants: Dark blue; Fungus: Dark green; Vertebrate; Purple.

**Figure 2 viruses-14-02810-f002:**
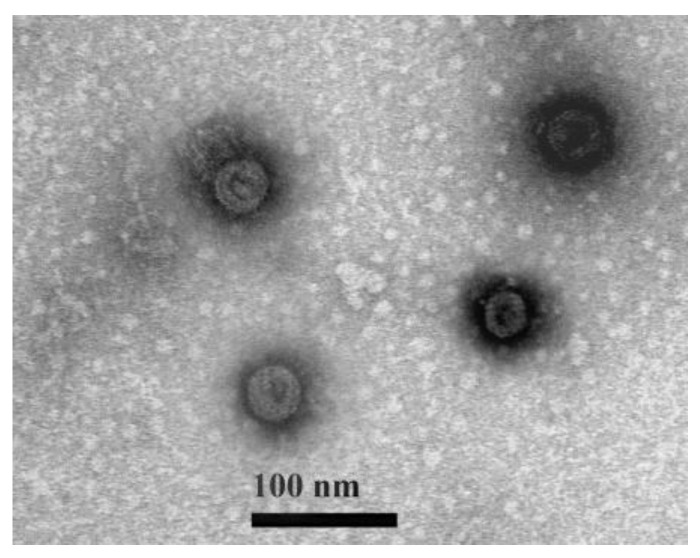
Electron micrographs of ARV virions.

**Figure 3 viruses-14-02810-f003:**
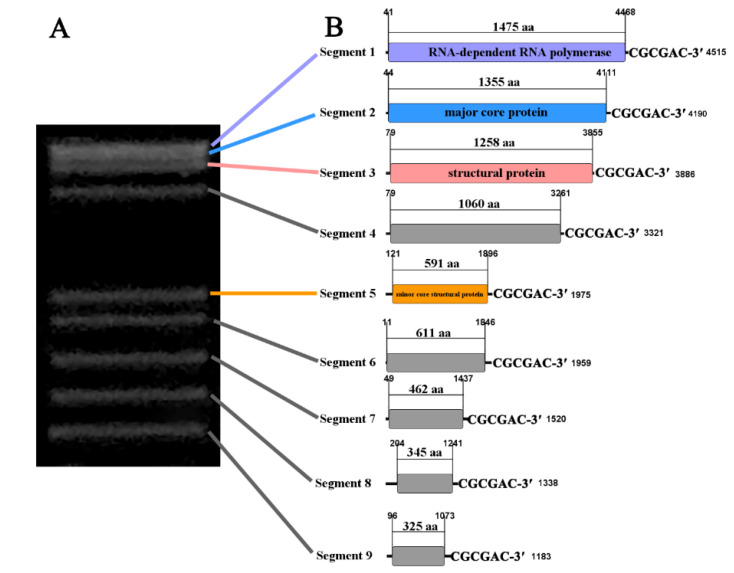
The complete genome of Acrididae reovirus (ARV). (**A**) Agarose gel electrophoresis of dsRNAs isolated from ARV virions. (**B**) Genomic organization of nine dsRNA segments of ARV. Five segments with no known homologs are indicated in grey.

**Figure 4 viruses-14-02810-f004:**
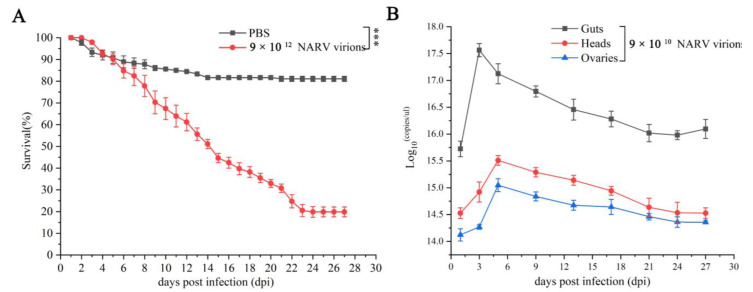
Survival and virus load of ARV-infected locusts. (**A**) The survival rate of ARV-infected locusts. Extremely significant differences were indicated by three asterisks (***, *p* < 0.001). Six replicates with twenty-five nymphs for each inoculation were generated. (**B**) The virus loads of the guts, heads, and ovaries at different time points. The viral load of a mixture of fifteen individuals was quantified at each time. Experiments were repeated five times.

**Figure 5 viruses-14-02810-f005:**
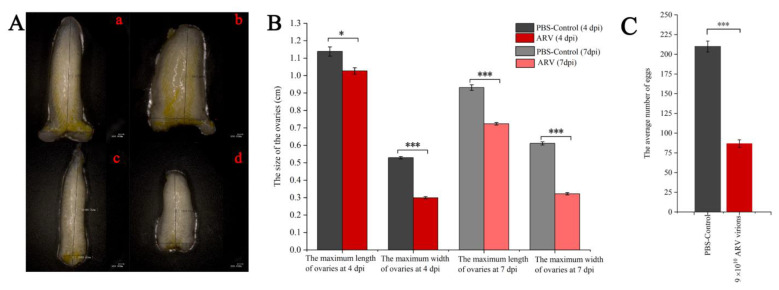
Comparison of ovarian morphology and the average number of eggs. (**A**) **a**: Adult’s ovary on fourth day of control group; **b**: Adult’s ovary on seventh day of control group; **c**: Adult’ ovary on fourth day of infection group; **d**: Adult’s ovary on seventh day of infection group. Fifteen ovaries were dissected at each time point, and this was repeated four times. (**B**) The size of the ovaries of uninfected and virus-infected adults. (**C**) The average number of eggs of uninfected and infected locusts. The egg-laying amount of fifteen pairs of locusts was recorded, and the experiments were repeated four times. Significant and extremely significant differences are indicated by one and three asterisks, respectively (*, *p* < 0.05; ***, *p* < 0.001).

**Table 1 viruses-14-02810-t001:** Viral load of infected nymphs and eggs.

Treatments	Viral Load (Virions/μL)	Time to Infection or Spawn (Days)
feces of infected nymphs	1.04 × 10^16^	3
nymphs exposed to the virus-laden feces	1.91 × 10^16^	3
eggs of infected adults	1.95 × 10^15^	1
clean eggs	0	1
nymphs hatched from cleaned eggs	0	1, 5, 9, 16, 23
nymphs hatched from uncleaned eggs	1.11 × 10^15^	1, 5, 9, 16, 23

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
