# Peer review of "Characterization of a Novel Pathogenic Reovirus in Grasshoppers"

_viruses, 2022, doi:10.3390/v14122810_

Round 1
Reviewer 1 Report
This study focused on a novel pathogenic reovirus in grasshoper. They determined that the complete genome of ARV which had 9 dsRNA segments. By injection, the proved that the ARV could be horizontally transmitted and replicated in Locusta migratoria. Viruses are usually harmful for the hosts and as resources of biopesticides, viruses benefit people on pest management. Interestingly, the ARV could decreased the fitness of Locusta migragoria by decreasing suvival rate and fecundity. Grasshoppers are major pests of crops all over the world. The pathogenic ARV could be potentially used to develop new biological strategy for controlling grasshopers.
Minor comments
1. Page5, Figure 1- It’s hard for reading the names in figure 1 and please change the color of some letters in this figure.
2. Page5, Figure 2- The diameters of these virons were diverse and some was no more than 40nm. Please re-describe the result.
3. Page6, the author proved that the ARV could be horizontally transmitted by injection and couldn’t be vertically tranmitted by transovarial. Could the ARV be horizontally transmitted by oral? Why did the author choose injection method for infection? About the transmission mode of ARV, the author should discuss or descibe more details.
4. Pages 6-7 (Figure 4 and Figure 5), about the bioassays, please added the repicated numbers for each experiment in the main text or figure legends.
5. Page 8, lines 283-293. This section shoule be re-write by citing references to discuss the interactions between insect-viruses and their insect hosts, especially reoviruses.
6. If possible, please concluded the study at the end of the manuscript.
Author Response
Dear Editor,
We would like to thank you for accepting our manuscript for publication pending minor revision. We would also like to thank two reviewers for the positive feedback and helpful comments.
We have revised the manuscript according to the comments made by both reviewers. Appended to this letter is our point-by-point response to the comments raised by the reviewers.
We very much hope the revised manuscript being published in Viruses.
Yours faithfully,
Chuan Cao and Yao Xu
Reviewer 1:
1.Page5, Figure 1- It’s hard for reading the names in figure 1 and please change the color of some letters in this figure.
Thank the reviewer for the suggestion. We have made corresponding changes in the latest manuscript.
Line 260-265: “Figure 1. Maximum likelihood phylogeny of Reoviridae. Phylogenetic tree was constructed using RdRp sequences. ARV was indicated by red arrow. The nine genera of the subfamily Spinareovirinae were marked in pink, and six genera of the subfamily Sedoreovirinae were marked in grey. Viruses were colored differently according to their hosts: Insects: Black; Plants: Dark blue; Fungus: Dark green; Vertebrate; Purple.”
2.Page5, Figure 2- The diameters of these virions were diverse and some was no more than 40nm. Please re-describe the result.
We have made corresponding changes in the latest manuscript.
Line 270-271: “ARV virions were similar in size to reported reoviruses, with an overall diameter of 40–50 nm (Figure 2).”
3.Page6, the author proved that the ARV could be horizontally transmitted by injection and couldn’t be vertically transmitted by transovarial. Could the ARV be horizontally transmitted by oral? Why did the author choose injection method for infection? About the transmission mode of ARV, the author should discuss or describe more details.
We chose injection rather than oral infection to better control the amount of virus infecting individual locusts. We’ve seen successful oral infection in a different pilot study, but it was not easy to quantitively measure how much virus was ingested by a nymph. We believe horizontal transmission is common for two reasons: 1) reovirus can be detected in the feces of infected locusts; 2) virus was detected in healthy that were nymphs exposed to the feces of infected nymphs. We also detect viruses on the surface of eggs laid by infected locusts and nymphs that hatched from these eggs. However, virus was not detected in nymphs that were hatched from sodium hypochlorite treated eggs. These results suggest potential transovarial transmission via egg surface. Thus, we have added description of the experiments in the methods part (line 215-221): “To reveal the transmission mode of ARV, twenty-five 3rd-instar nymphs were exposed to the feces of virus-infected locusts for 2 hours, and the viral load of the locusts was measured after 24 hours. Additionally, eggs laid by infection adults were collected. Total RNA was extracted from two types of eggs, including those that had not been cleaned and had been disinfected with sodium hypochlorite. Viral loads were also measured for nymphs hatched from cleaned or not cleaned eggs. qPCR was used to test for viral titers of eggs. Experiments were repeated 5 times.” We then modified in the result part (line 345-352) to: “In addition, ARV can be detected from the feces and from nymphs exposed to the feces of infected nymphs (Table1). This result indicated that ARV could be horizontally transmitted by feces of infected locusts. ARV can also be detected on the surface of eggs and in nymphs that hatched from these eggs (Table 1). However, provided the egg surface was disinfected using compound sodium hypochlorite disinfectant, no transovarial transmission has been observed (Table1). ARV was not detected in nymphs that were hatched from sodium hypochlorite treated eggs. These results suggest potential transovarial transmission via egg surface.”
Table 1. viral load of infected nymphs and eggs.
|
Treatments |
viral load (virions/ul) |
Time to infection or spawn (days) |
|
feces of infected nymphs |
1.04 â…¹10 16 |
3 |
|
nymphs exposed to the virus-laden feces |
1.91 â…¹10 16 |
3 |
|
eggs of infected adults |
1.95 â…¹10 15 |
1 |
|
clean eggs |
0 |
1 |
|
nymphs hatched from cleaned eggs |
0 |
1, 5, 9, 16, 23 |
|
nymphs hatched from uncleaned eggs |
1.11â…¹10 15 |
1, 5, 9, 16, 23 |
- Pages 6-7 (Figure 4 and Figure 5), about the bioassays, please added the repeated numbers for each experiment in the main text or figure legends.
Thank the reviewer for the suggestion. We have made corresponding changes in the manuscript.
Line 322-325: “Figure 4. Survival and virus load of ARV-infected locusts. (A) The survival rate of ARV-infected locusts. Six replicates with twenty-five nymphs for each inoculation were generated. (B) The virus loads of the guts, heads and ovaries at different time points. The viral load of a mixture of fifteen individuals was quantified at each time. Experiments were repeated five times.”
Line 335-341: “Figure 5. Comparison of ovarian morphology and the average number of eggs. (A) a: Adult’s ovary on fourth day of control group; b: Adult’s ovary on seventh day of control group; c: Adult’ ovary on fourth day of infection group; d: Adult’s ovary on seventh day of infection group. Fifteen ovaries were dissected each time, and this was repeated four times. (B) The average number of eggs of uninfected and infected locusts. The egg laying amount of fifteen pairs of locusts was recorded, and the experiments were repeated four times.”
5.Page 8, lines 283-293. This section should be rewrite by citing references to discuss the interactions between insect-viruses and their insect hosts, especially reoviruses.
Thank the reviewer for the suggestion. We have made corresponding changes in the manuscript line 379-398.
“Study of virus pathogenicity and its interaction with hosts is often limited by the difficulty of virus isolation and culturing. There are limited studies about the interaction between reoviruses and their hosts. For example, Diaphorina citri reovirus (DcRV) is recently identified from Diaphorina citri populations and considered a novel species of Reoviridae [27]. Notably, DcRV-encoded non-structural protein P10 is associated with virus spread and biparental transmission to progeny [28]. Bombyx mori cytoplasmic polyhedrosis virus (BmCPV) is a relatively well studied reovirus. A recent study has discovered that ROS induced by vSP27 can suppress BmCPV infection by the NF-κB signaling pathway. vSP27 is translated from a BmCPV derived circular RNA (circRNA-vSP27) [29]. Fortunately, we are able to generate a single reovirus clone which actively replicate in insect cells and locusts. It provides an excellent laboratory replication model for studying virus-host interactions.”
- If possible, please concluded the study at the end of the manuscript.
We have concluded the study at the end of the manuscript.
Line 411-419: “5. Conclusions
In this work, we identified and isolated a novel reovirus from grasshoppers, named Acrididae reovirus (ARV). ARV can be successfully cultured in insect cell cultures and live locusts. By genome sequencing, our results showed that the complete genome of ARV comprised of nine segments. All genomic segments of ARV shared a conserved hexanucleotide terminus at their 3’ ends (CGCGAC-3′), which was different from previously reported reoviruses. Importantly, ARV was pathogenic to its host by reducing host survival and reproduction. These results indicated that ARV can be a good candidate for biological control of grasshopper pests.”

Reviewer 2 Report
The paper by Xu, Cao and colleagues is interesting and well written, and presents an intriguing link between their newly discovered Reovirus, ARV, and potentially promoting the development of new biological control agents in Grasshoppers. While I feel several steps need to be taken to more thoroughly link ARV as the novel pathogenic reovirus in the grasshoppers. The manuscript would be improved if the following concerns were addressed:
1. At a minimum, the authors need to run RT-PCR on RNA extracted from multiple organs from affected and unaffected Grasshoppers to demonstrate the presence of the virus within the tissues. As far as I can tell, there are no data showing the presence of the virus in affected grasshoppers, only from the cell line. A more powerful and convincing approach, would be to perform qPCR, so that the viral burden can be assessed and correlated with disease.
2. The Viruses readership (and this reviewer!) is not generally familiar with grasshopper pathology, so some effort to bridge the gap would be appreciated. This could include comparator pictures of unaffected grasshopper and cartoons that clearly illustrate out the clinical features. I would like to see a summary table of clinical signs observed in "naturally" morbid vs actively infected grasshopper.
3. Can the authors provide more details on the outbreak? Were juveniles or ovaries more/less affected? What is the relevance of the temperature? Please concisely reproduce some of the most relevant details for context, rather than showing a single picture of sizes of ovary in diseased grasshopper.
4. Furthermore, the authors need to supply more information about the animal experiments: How was the grasshopper tested prior to the experiments? How many grasshoppers were used in the animal experiments? The grasshoppers were so called “healthy” grasshopper, Were they tested and found free for common grasshopper pathogens?
5. The experimentally challenged groups of grasshoppers were injected with virus and there were no cohabitation experiments performed. Cohabitation reflects better the natural way of transmission, while injection does not. By injecting the material, you skip the possible influence of mucosal immunity. Therefore, the authors should state a reservation in the text due to the use of injection in the challenge experiments.
6. ARV from tissues of diseased grasshopper by using the transmission electron microscopy observation should be provided, rather than the just ARV virions harvested from cell culture.
7. Line 131-133: “A classical TCID50 assays failed to titrate the viral particles because ARV did not induce cytopathic effect in insect cell cultures. Quantitative Real-Time PCR (qRT-PCR) methodology had been proved effective in the determination of reovirus stock titers”. Line 191-192: “A single clone of ARV was further cultured for 7 days to produce pure ARV”. Did these two descriptions contradict each other? How to figure out the viral single clone without CPE?
8. More accurate divisions in the materials and methods section should be provided.
e.g. section 2.2: Please put the contents of viral RNA extraction and qPCR in Virus purification section into the section 2.5 (Virus quantification) or separate these contents as an independent section.
9. Did the authors not include a statistical analysis in materials and methods section? A big concern in this study.
10. Lines 88 and 131: should be 1 x 104 cells and TCID50? Please figure out them.
Author Response
Dear Editor,
We would like to thank you for accepting our manuscript for publication pending minor revision. We would also like to thank two reviewers for the positive feedback and helpful comments.
We have revised the manuscript according to the comments made by both reviewers. Appended to this letter is our point-by-point response to the comments raised by the reviewers.
We very much hope the revised manuscript being published in Viruses.
Yours faithfully,
Chuan Cao and Yao Xu
Reviewer 2:
- At a minimum, the authors need to run RT-PCR on RNA extracted from multiple organs from affected and unaffected Grasshoppers to demonstrate the presence of the virus within the tissues. As far as I can tell, there are no data showing the presence of the virus in affected grasshoppers, only from the cell line. A more powerful and convincing approach, would be to perform qPCR, so that the viral burden can be assessed and correlated with disease.
Thank the reviewer for the suggestion. The qPCR method has been used to demonstrate the successful infection of the reovirus in locust tissues such as heads, guts and ovaries. Admittedly, we did not describe in detail in the Methods section. We have made corresponding changes in the manuscript. Line 201-203, 207-210 “Survival of inoculated locusts was recorded daily until no deaths occur. The qPCR method was used to demonstrate the successful infection of the reovirus in locusts. Six replicates with twenty-five nymphs for each inoculation were generated.” “RT-PCR on RNA extracted of guts, ovaries and heads from infected and uninfected locusts was carried out. qPCR was used to test for viral titers. The viral load of a mixture of fifteen individuals was quantified at each time point. Experiments were repeated five times.”
- The Viruses readership is not generally familiar with grasshopper pathology, so some effort to bridge the gap would be appreciated. This could include comparator pictures of unaffected grasshopper and cartoons that clearly illustrate out the clinical features. I would like to see a summary table of clinical signs observed in "naturally" morbid vs actively infected grasshopper.
Thank the reviewer for the suggestion. We found no obvious pathological features other than the observed death following high doses of infection. Infection with low doses of reovirus may affect the growth, development and feeding of locusts. Unfortunately, we did not investigate further in the present study. This will be the focus in future research on viral pathogenic mechanisms.
- Can the authors provide more details on the outbreak? Were juveniles or ovaries more/less affected? What is the relevance of the temperature? Please concisely reproduce some of the most relevant details for context, rather than showing a single picture of sizes of ovary in diseased grasshopper.
Thank the reviewer for the suggestion. We think it would be interesting to carry out more experiments further exploring different factors that affect infections. However, in the present study we did observe consistent smaller ovaries and fewer egg production for the infected individuals.
Figure 5. Comparison of ovarian morphology and the average number of eggs. (A) a: Adult’s ovary on fourth day of control group; b: Adult’s ovary on seventh day of control group; c: Adult’ ovary on fourth day of infection group; d: Adult’s ovary on seventh day of infection group. Fifteen ovaries were dissected each time, and this was repeated four times. (B) The size of the ovaries of uninfected and virus-infected adults. (C) The average number of eggs of uninfected and infected locusts. The egg laying amount of fifteen pairs of locusts was recorded, and the experiments were repeated four times.
- Furthermore, the authors need to supply more information about the animal experiments: How was the grasshopper tested prior to the experiments? How many grasshoppers were used in the animal experiments? The grasshoppers were so called “healthy” grasshopper, Were they tested and found free for common grasshopper pathogens?
We have made corresponding changes in the manuscript.
Line 73-74, 203, 209-210: “All samples in this study were cultured in a dedicated incubator to avoid the infection of other pathogens.” For infectivity assays, “Six replicates with twenty-five nymphs for each inoculation were generated.” “The viral load of a mixture of fifteen individuals was quantified at each time point. Experiments were repeated five times.”
Eggs were soaked in sodium hypochlorite for 10 minutes before hatching. Most pathogens, including viruses, bacteria and fungi, will be removed from the egg surface.
- The experimentally challenged groups of grasshoppers were injected with virus and there were no cohabitation experiments performed. Cohabitation reflects better the natural way of transmission, while injection does not. By injecting the material, you skip the possible influence of mucosal immunity. Therefore, the authors should state a reservation in the text due to the use of injection in the challenge experiments.
We chose injection rather than oral infection to better control the amount of virus infecting individual locusts. We’ve seen successful oral infection in a different pilot study, but it was not easy to quantitively measure how much virus was ingested by a nymph.
- ARV from tissues of diseased grasshopper by using the transmission electron microscopy observation should be provided, rather than the just ARV virions harvested from cell culture.
Thank the reviewer for the suggestion. We showed the EM picture of ARV generated from cell culture as we could produce pure ARV virions using cell culture. However, we did not perform the EM experiment of infected tissues in the present study. However, we are convinced of infection in various tissues by qRT-PCR.
- Line 131-133: “A classical TCID50 assays failed to titrate the viral particles because ARV did not induce cytopathic effect in insect cell cultures. Quantitative Real-Time PCR (qRT-PCR) methodology had been proved effective in the determination of reovirus stock titers”. Line 191-192: “A single clone of ARV was further cultured for 7 days to produce pure ARV”. Did these two descriptions contradict each other? How to figure out the viral single clone without CPE?
Real-time fluorescence quantitative PCR (qPCR) was used to figure out the viral single clone without CPE. Please also refer to the description of our method (line 97-100). “Real-time fluorescence quantitative PCR (qPCR) was used to detect reovirus and other viruses. Specific primers for different viruses were listed in our previous study (Table S1) [10]. Only reovirus-infected cells were kept to further expand.”
- More accurate divisions in the materials and methods section should be provided.
e.g. section 2.2: Please put the contents of viral RNA extraction and qPCR in Virus purification section into the section 2.5 (Virus quantification) or
Thank the reviewer for the suggestion. We have separated these contents as an independent section (line 131-137).
- Did the authors not include a statistical analysis in materials and methods section? A big concern in this study.
We have included the statistical analysis in the text.
Line 222-227: “2.8. Statistical Analysis
To compare the qualitative and quantitative variables between groups, the Student’s T-test or the Mann-Whitney U test were used in a univariate analysis. The results were considered statistically significant at p < 0.05; specific p-values were provided in the text. Statistical analysis was performed using IBM SPSS Statistics 26.0 (Chicago, IL, USA) and GraphPad Prism 8.0.1(La Jolla, CA, USA).”
- Lines 88 and 131: should be 1 x 104 cells and TCID50? Please figure out them.
We have made corresponding changes in the manuscript. 1 x 104 cells changed to 1 x 104 cells (line 90), TCID50 changed to TCID50 (line 167).
